# Learning A Risk-Aware Trajectory Planner From Demonstrations Using Logic Monitor

**Xiao Li**
MIT
xiaoli@mit.edu

**Jonathan A. DeCastro**
Toyota Research Institute
jonathan.decastro@tri.global

**Cristian-Ioan Vasile**
Lehigh University
cvasile@lehigh.edu

**Sertac Karaman**
MIT
sertac@mit.edu

**Daniela Rus**
MIT
rus@mit.edu

**Abstract:** Risk awareness is an important factor to consider when deploying policies on robots in the real-world. Defining the right set of risk metrics can be difficult. In this work, we use a logic monitor that keeps track of the environmental agents' behaviors and provides a risk metric that the controlled agent can incorporate during planning. We introduce LogicRiskNet, a learning structure that can be constructed from temporal logic formulas describing rules governing a safe agent's behaviors. The network's parameters can be learned from demonstration data. By using temporal logic, the network provides an interpretable architecture that can explain what risk metrics are important to the human. We integrate LogicRiskNet in an inverse optimal control (IOC) framework and show that we can learn to generate trajectory plans that accurately mimic the expert's risk handling behaviors solely from demonstration data. We evaluate our method on a real-world driving dataset.

**Keywords:** Learning from demonstrations, Temporal Logic, Autonomous Driving

## 1 Introduction

Imagine a driver that is trying to turn across traffic at an unprotected intersection where there is oncoming traffic. When faced with the situation shown in Figure 1(a), the driver needs to decide when it is safe to make the turn. The driver makes this decision by evaluating how risky the nearest oncoming vehicle is in terms of the intended turning maneuver. The risk may be dependent on a number of factors, including how far the vehicle is from the intersection, how fast it is driving and whether it has the intention to slow down. If we wish to incorporate such risk-aware decision making into an autonomous driving system, we are faced with two problems: (1) how do we accurately model the risk factors that humans take into account while driving, and (2) how do we generate plans from these risk factors. With the development of data-driven autonomous driving, we have access to a growing amount of large-scale human driving data. Large-scale data contain not only normal driving behaviors, but also what drivers do in order to manage risk. Taking advantage of these data, our goal in this work is to *develop a risk representation that can expressively describe the desired/undesired behaviors of road agents and with parameters learnable from demonstration data.* We aim to obtain risk metrics that reflect that of the human demonstrator, and generate risk-aware trajectory plans using these metrics.

Recent work on risk and uncertainty aware policy learning aim to incorporate common risk constructs such as conditional value at risk, worst-case risk, etc., into the problem formulation either as an auxiliary loss or constraints [1, 2]. Many then use reinforcement learning (RL) to obtain the policy. The main limitation of applying RL to autonomous driving is its exploration requirement. It can be difficult to explore safely especially when we wish to deal with risky scenarios. Work on uncertainty-aware imitation learning [3] and risk-aware offline RL [4] address the exploration problem, but are limited in the types of risks they can express.

5th Conference on Robot Learning (CoRL 2021), London, UK.

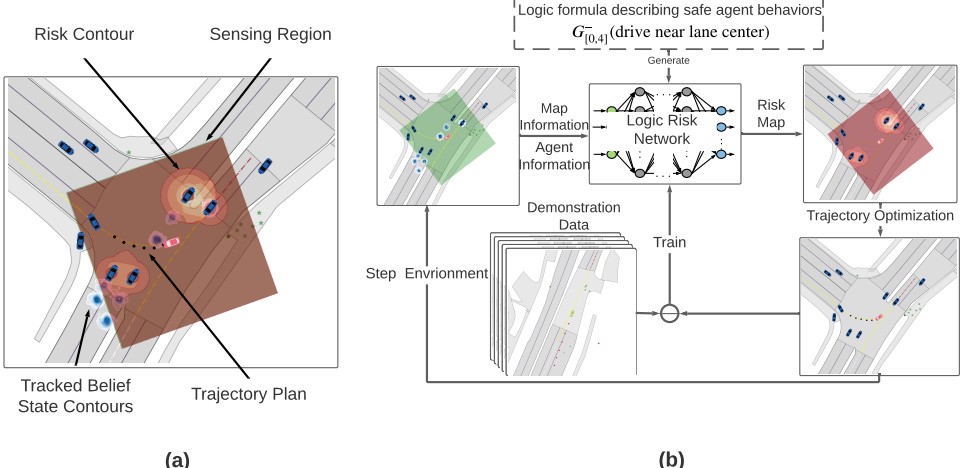

**Figure 1: LogicRiskNet-IOC framework.** (**a**) A pictorial depiction of our setup. The red car is the ego vehicle (controlled by our planner). The blue cars move according to the trajectories recorded in the dataset. Their trailing contours represent the tracking distributions. The red contours around the ado vehicles (neighboring vehicles not controlled by us) present their risks (the brighter the contour the higher the risk.) (**b**) An illustration of our inverse optimal control (IOC) procedure. The outer loop generates trajectory plans according to the risk map. The inner loop updates LogicRiskNet's parameters using a kind of feature matching.

It is important to be risk-averse in safety critical applications. It is often more important to be averse to the right set of risks. In this paper, we want to answer the question "if we have observed our nearby road agents for a while, can we develop a risk model that quantifies how risky they are based on their past behaviors in the same way that humans would." We take inspiration from real-time verification [5] and formal synthesis [6] and introduce a differentiable logic monitor which we refer to as *LogicRiskNet* as an expressive and learnable risk representation. To summarize our contributions, we

- introduce a differentiable logic-based risk metric and show that this metric is able to describe complex, temporally-extended risk events as logic formulas and learn its parameters from human driving data;

- show that we can not only use LogicRiskNet to generate risk-aware trajectories, given its hierarchical monitoring properties, we can also use it to explain *why* an agent is risky;

- demonstrate in a real-world driving dataset that our method yields controllable risk-averse or, alternatively, risk-seeking behaviors.

Note that, while we apply our approach to the domain of autonomous driving, we can equally-well extend our approach to other domains, such as robotics.

## 2 Background: Past Time Signal Temporal Logic (PtSTL)

STL [7] offers a formalism for expressing and reasoning among a rich set of rules. The rules are defined over predicates (inequalities of real-valued functions). In standard STL, the formulas are forward-looking, meaning that the formulas are evaluated by looking at future trajectories. In this work, we need to evaluate formulas using past trajectories (trajectories of road agents we have tracked over the past), using this past information to compute a risk of violating these rules in the future. Therefore, our rules will be encoded as a set of *past time STL* (ptSTL) [8] formulas with the following syntax

$$\phi := p(x) < \epsilon \,|\, \neg\phi \,|\, \phi \wedge \phi \,|\, \phi \vee \phi \,|\, F_{[a,b)}^{-}\phi \,|\, G_{[a,b)}^{-}\phi, \tag{1}$$

where $a, b \in \mathbb{R}_{\geq 0}$, $a < b$, represent finite time bounds; $\phi$ is a ptSTL formula; $p(x) < \epsilon$ is a predicate; $x \in X \subseteq \mathbb{R}^n$ is a state; $p(\cdot) : \mathbb{R}^n \to \mathbb{R}$ is the predicate function, $\epsilon \in \mathbb{R}$ is a constant. $\neg$ (not), $\wedge$ (and) and $\vee$ (or) are Boolean operators. $F^-$ (eventually) and $G^-$ (always) are past time temporal operators. Here $F^-_{[a,b)}\phi$ requires $\phi$ to be true at least once in the time interval $t \in (-b, -a]$ whereas $G^-_{[a,b)}\phi$ requires $\phi$ to be true for all of this interval.

Let $x^{t_0:t_1} = \{x_{t_0}, ..., x_{t_1}\}$ denote the state trajectory from time $t_0$ to $t_1$. $x^{t_0:t_1} \models \phi$ denotes that trajectory $x^{t_0:t_1}$ satisfies $\phi$ (i.e. $\phi$ evaluates to true under $x^{t_0:t_1}$). The Boolean semantics of ptSTL are provided in Appendix A.

An example ptSTL formula is $F^-_{[\tau_0,\tau_1)}(|x| < 2) \wedge G^-_{[\tau_1,\tau_2)}(x > 3)$ which means that "$|x_t| < 2$ is true for at least one $t \in (-\tau_1, -\tau_0]$ and $x_t > 3$ is true for all $t \in (-\tau_2, -\tau_1]$".

Similar to STL, ptSTL is equipped with a *robustness degree* (or robustness for short) that quantifies the level of satisfaction of a trajectory with respective to a formula. A robustness function, denoted $r$, takes in a trajectory and a ptSTL formula and outputs a real-valued number. A robustness greater than zero signifies that the trajectory satisfies the given formula, i.e. $r(x^{t_0,t_1}, \phi) > 0 \Rightarrow x^{t_0,t_1} \models \phi$. Negative robustness implies violation of the formula. The formal definition of robustness is also given in the Appendix A.

Note that each predicate relationship $p(x) < \epsilon$ appearing in $\phi$ has parameters that can be fit to how humans behave. For example, the parameterized ptSTL formula $\phi = G^-_{[\tau_0,\tau_1]}(\alpha_1 x + \gamma_1 y < \epsilon_1) \wedge F_{[\tau_2,\tau_3]}(\alpha_2 x^2 > \epsilon_2)$ has parameters $\xi = \{(\alpha_1, \gamma_1), \epsilon_1, \alpha_2, \epsilon_2\}$. Hence, we parameterize each predicate appearing in a formula $\phi$, denoting the parameterized formula $\phi^\xi$ (see, e.g., [9]).

## 3 Problem Formulation and Approach

Let $\boldsymbol{x} = (\boldsymbol{p}, \psi, v)$ denote the state of a vehicle that includes position (x-y coordinates), heading and velocity (which we later use in a unicycle model). We call the vehicle that is controlled by our planner the *ego* agent, and all other traffic participants not controlled by us *ado* agents. Denote $\Omega_T$ as the set of trajectories with horizon $T$ (i.e. $\boldsymbol{\omega}_T = \boldsymbol{x}^{t:t+T} \in \Omega_T$). Suppose that the ego vehicle tracks its nearest $N$ ado agents as well as itself via a tracking algorithm (such as a Kalman filter). This algorithm returns a set of belief states at time step $t$, $h_t = \left[\boldsymbol{x}_e^t, \boldsymbol{\Sigma}_e^t, \boldsymbol{x}_{a_i}^t, \boldsymbol{\Sigma}_{a_i}^t\right]$, $i \in [0, N)$, which are concatenated into a history of length $H$ as $\boldsymbol{h}_H^t = [h_{t-H}, \ldots, h_t]$. Here, $\boldsymbol{x}_{e,a_i}^t$ and $\boldsymbol{\Sigma}_{e,a_i}^t$ are the mean values and covariances for the ego and $i$-th ado vehicle states. $\mathcal{H}_H$ is the set of such histories (i.e. $\boldsymbol{h}_H^t \in \mathcal{H}_H$). Define a *dataset* $\mathcal{D} = [\Omega_T, U_T, \mathcal{H}_H]$ where $\mathcal{H}_H$ is obtained from sensors, $\Omega_T$ is the ego vehicle's ground truth future trajectory, $\boldsymbol{u}_T = \left[\boldsymbol{u}^t, ..., \boldsymbol{u}^{t+T-1}\right] \in U_T$ is the corresponding sequence of controls (speed and steering). Let *feature* $f : \Omega_T \times U_T \to \mathbb{R}$ be a function that maps a trajectory of states and controls to a real value (e.g. average distance to the center lane). $\boldsymbol{f}(\boldsymbol{\omega}_T) = \left(f^1(\boldsymbol{\omega}_T), ..., f^n(\boldsymbol{\omega}_T)\right) \in \mathbb{R}^n$ is a vector of such features. Our problem is formally defined as follows

**Problem 1.** *Given (1) a parametric ptSTL formula $\phi^\xi$ that defines the behavior of a safe ado agent, (2) a feature $f^{\phi^\xi}$ representing ado agent risks, (3) a set of user defined features representing behaviors of the ego agent $\boldsymbol{f}$, (4) a demonstrations dataset $\mathcal{D}$ that contains a human driver's decisions and environment information, find a trajectory distribution $\mathcal{P}^\theta : \Omega_T \to [0, 1]$ parametrized by $\theta$ ($\theta$ contains the ptSTL parameters $\xi$ as well as feature combination weights) that best matches human behavior; i.e., minimizes the following objective functions*

$$J_1 = \mathbb{E}_{\mathcal{P}^\theta}\left[\boldsymbol{f}(\tilde{\boldsymbol{\omega}}_T)\right] - \boldsymbol{f}(\boldsymbol{\omega}_T) \text{ and } J_2 = \mathbb{E}_{\mathcal{P}^\theta}\left[f^{\phi^\xi}(\tilde{\boldsymbol{\omega}}_T)\right] - f^{\phi^\xi}(\boldsymbol{\omega}_T) \quad (2)$$

*where $\boldsymbol{\omega}_T$ is a trajectory from the demonstration dataset. $\tilde{\boldsymbol{\omega}}_T$ is a trajectory sampled from $\mathcal{P}^\theta$.*

Problem 1 defines a feature matching problem similar to many inverse reinforcement learning (IRL) [10] and IOC [11] formulations. We design a set of features $\boldsymbol{f}$ to capture driving preferences in non-risky situations, but also include a risk-based feature $f^{\phi^\xi}$ capturing the risk management preferences seen in demonstrations. From the demonstration data we then learn the risk metric parameters $\xi$ and the weights that combine the feature values. This combination gives us a cost model that, when solved, yields driving behaviors similar to the demonstrator (in the context of our defined

features and risk models). While existing IOC approaches are able to imitate standard driving behaviors such that the generated trajectories from the cost model are exponentially more preferred by the agent (under maximum-entropy formulations, e.g. [11]). In our case, the additional risk feature provides the capacity to generalize our model better under risky situations without hindering the ability to replicate the driving style of human demonstrators in normal scenarios.

## 4 Logic Monitor Guided Risk-Aware LfD

In this section, we will describe in detail our risk-aware LfD framework. We will first introduce the *LogicRiskNet* - a differentiable syntax tree generated from a parametric ptSTL formula. The network takes as input the tracking histories $\boldsymbol{h}_T$ and outputs a risk assignment for each tracked vehicle based on their past behaviors. This risk assignment can be used to monitor which vehicles are risky in a given scene. We then show how LogicRiskNet can be incorporated into an IOC framework that is able to learn this syntax tree from demonstrations while simultaneously learn a cost function that specifies a risk-aware planning problem, from which a trajectory can be solved efficiently.

Our method is depicted in Figure 1(b) where there are 2 nested loops. The outer loop starts with the map and agent information which are passed through the LogicRiskNet to construct a risk map. This risk map constitutes the objective function for a trajectory planner. The generated trajectory serves to control the ego agent in a receding horizon fashion. In the inner loop, the same set optimized trajectories are also compared against human demonstrated trajectories in the same situations, and their differences serve as training signals to the LogicRiskNet.

**LogicRiskNet - a differentiable parametric ptSTL risk monitor.** We describe how we may construct a ptSTL risk monitor based on a learned stochastic risk measure that can be applied to a probabilistic description of human behaviors. We extend the definition of robustness degree (introduced in Section 2) to encompass belief states. In order to calculate the risk of a trajectory of belief states w.r.t a ptSTL formula $\phi$, we need to modify the definition of the robustness. Recall that the basic element of $\phi$ is a predicate of the form $p(x) < c$, define $\alpha(x) = c - p(x)$ as the predicate (this is same as the robustness degree for the predicate). Since $x$ is stochastic, rather than deterministic, we can evaluate ptSTL formulas instead using a risk measure $\rho : \mathcal{X} \to \mathbb{R}$, representing expectation, mean-variance, value-at-risk, etc. [1] In this work, we assume the expectation risk measure, but can equally well replace this with others. We can then apply this risk measure to the robustness definition yielding $\alpha^\rho(x) = \rho(\alpha(x))$, $x \in \mathcal{X}$ (This treatment is similar to [13]).

Given a parametric ptSTL formula $\phi^\xi$, a risk measure $\rho(\cdot)$ and the tracking history $\boldsymbol{h}_H$, we can construct the *robustness risk* for $\phi^\xi$ as

$$R^\phi(\boldsymbol{h}_H \mid \xi) \;=\; r^\rho\left(\boldsymbol{h}_H, \phi^\xi\right) \tag{3}$$

where $r^\rho\left(\boldsymbol{h}_H, \phi^\xi\right)$ is the robustness applied to the risk-based predicate $\alpha^\rho$. Once the risk predicate values are calculated at each time step in the trajectory, the robustness calculations for Boolean and temporal operators can be carried out. In order to tune the parameters $\xi$ using demonstration data, we follow the approach in [14]. Specifically, if we replace the max and min functions in the robustness definition with softmax approximations, we can construct a computation graph from $R^\phi$ with the predicate parameters at the leaves and the robustness risk at the root. This makes it possible to learn $\xi$ using backpropagation. The intuition behind Equation 3 is that given a ptSTL formula $\phi$ parameterized by $\xi$, agents' past observations $\boldsymbol{h}_H$, and a risk measure $\rho$, the robustness risk $R^\phi$ can be calculated to represent agent risk levels in terms of satisfaction of $\phi$.We provide a step-by-step example of constructing the LogicRiskNet in Appendix F.

**LfD using LogicRiskNet.** With LogicRiskNet $R^\phi$, we are able to characterize risky ado vehicles as using the following risk feature:

$$f^\phi(\boldsymbol{h}_H, \boldsymbol{\omega}_T \mid \xi) = \sum_{i=0}^{N-1} \sum_{t=0}^{T-1} -R^\phi(\boldsymbol{h}_H^i \mid \xi) \frac{1}{||\boldsymbol{p}_e^t - \boldsymbol{p}_{a_i}^t||^2} \tag{4}$$

---

[1] We further require that the risk measure is *coherent*; see [12].

---

**Algorithm 1** Trajectory Optimization

---
1: $\boldsymbol{Inputs}$: initial controls $\tilde{\boldsymbol{u}}_T^{init}$; initial ego vehicle state $\boldsymbol{x}_e^0$; tracking history $\boldsymbol{h}_H$; lane informa-
   tion $\mathcal{T}$; control effort weight $\beta^{\boldsymbol{u}}$; lane tracking weight $\beta^{\mathcal{N}}$; ptSTL formula parameters $\xi$;
2: $\tilde{\boldsymbol{u}}_T \leftarrow \tilde{\boldsymbol{u}}_T^{init}$
3: **for** i=1 ... N **do**
4:    $\tilde{\boldsymbol{\omega}}_e \leftarrow \texttt{ConstructTrajectory}(\tilde{\boldsymbol{u}}_T, \boldsymbol{x}_e^0)$
5:    $\tilde{f}^\phi \leftarrow \tilde{f}^\phi\left(\boldsymbol{h}_H, \tilde{\boldsymbol{\omega}}_e \mid \xi\right), \tilde{f}^{\boldsymbol{u}} \leftarrow \tilde{f}^{\boldsymbol{u}}\left(\tilde{\boldsymbol{u}_T}\right), \tilde{f}^{\mathcal{N}} \leftarrow \tilde{f}^{\mathcal{N}}\left(\tilde{\boldsymbol{\omega}}_e, \mathcal{N}_T\right)$    ▷ construct features
6:    $\tilde{L} = \tilde{f}^\phi + \beta^{\boldsymbol{u}} \cdot \tilde{f}^{\boldsymbol{u}} + \beta^{\mathcal{N}} \cdot \tilde{f}^{\mathcal{N}}$    ▷ construct objective
7:    $\tilde{\boldsymbol{u}_T} \leftarrow \tilde{\boldsymbol{u}_T} - \tilde{\gamma}\nabla_{\boldsymbol{u}_T}\tilde{L}$    ▷ update control sequence
8: **end for**
9: $\tilde{\boldsymbol{\omega}}_e \leftarrow \texttt{ConstructTrajectory}\left(\tilde{\boldsymbol{u}}_T, \boldsymbol{x}_e^0\right)$
10: **return** $\tilde{\boldsymbol{\omega}}_e, \tilde{f}^\phi, \tilde{f}^{\boldsymbol{u}}, \tilde{f}^{\mathcal{N}}$

---

where $\boldsymbol{p}_e^t$ and $\boldsymbol{p}_{a_i}^t$, $t \in [0, T)$ are the ego and ado vehicles' positions at time $t$. $\boldsymbol{h}_H^i$ is the tracking history of ado vehicle $i$ along with the history of the ego agent. Essentially, (4) describes a collision feature scaled by the risk $R^\phi$, where the inverse-squared distance may be viewed as a generalization of potential fields [15]. A safe trajectory should yield a low $f^\phi$ value. A positive $R^\phi$ signifies a rule abiding ado agent whereas a negative $R^\phi$ signifies a rule-violating agent. Intuitively, we use the past observations to determine how risky an ado agent is, and by solving for trajectories that minimize $f^\phi$ we can control the ego agent to stay away from risky ado agents. Our potential-field formulation of risk acts to approximate the behavior of drivers in reacting to potential collisions but do not over-react if collision is not imminent.

To imitate risk-free driving behavior, we additionally introduce a control effort feature and a nominal trajectory feature for the ego vehicle as follows

$$f^{\boldsymbol{u}}(\boldsymbol{u}_T) = \sum_{t=0}^{T-1} ||\boldsymbol{u}^t||^2 \text{ where } \boldsymbol{u} = (v, \dot{\psi}), \quad f^{\mathcal{N}}(\boldsymbol{\omega}_T, \mathcal{N}_T) = \sum_{t=0}^{T-1} ||\boldsymbol{p}_e^t - \boldsymbol{p}_{\mathcal{N}}^t||^2 \quad (5)$$

where $\boldsymbol{p}_{\mathcal{N}}^t \in \mathcal{N}_T$ are points on the nominal trajectory. Nominal trajectories can be the human ego agent's trajectories (during training) or trajectories on the target lane center (during deployment). These features encourage the ego agent to follow a trajectory (and speed) profile with smooth control efforts. Given features $f^\phi, f^{\boldsymbol{u}}, f^{\mathcal{N}}$, we solve for a trajectory plan $\boldsymbol{\omega}_T$ by minimizing $f^\phi + \beta^{\boldsymbol{u}} \cdot f^{\boldsymbol{u}} + \beta^{\mathcal{N}} \cdot f^{\mathcal{N}}$ using gradient descent. Algorithm 1 describes this trajectory optimization process.

In Algorithm 1, line 4 describes the process of using a unicycle model to construct a trajectory from a control sequence. This procedure allows us to generate kinematically feasible trajectories compared to direct trajectory optimization. In calculating the risk feature $f^\phi$, we use a constant velocity and heading model estimate of ado agents' future trajectories $\boldsymbol{p}_{a_i}^t, t \in [0, T-1)$. During training, the nominal trajectory $\mathcal{N}_T$ is the ego vehicle's ground truth future trajectory from the dataset. During deployment, this nominal trajectory is taken from points on the target center lane which can be obtained from route planning.

Algorithm 2 describes how the parameters from the LogicRiskNet along with the objective function weights are learned from demonstrations using a maximum entropy model of uncertainty.

All entities with superscript $d$ are taken from the demonstrations. Line 4 calculates the sum of features obtained from trajectories from the human driver averaged over the dataset. Lines 6-10 calculates the same but with trajectories obtained from Algorithm 1. Line 12 updates the parameters such that $A_{f_{\mathcal{D}}^\phi}$ and $A_{\tilde{f}^\phi}$ can match.

## 5   Experimental Results

We train and evaluate our method on the NuScenes dataset. NuScenes [16] is a dataset for autonomous driving based in Boston and Singapore. It contains 850 scenes each 20s long, containing 23 object classes and HD semantic maps with 11 annotated layers. We chose this particular dataset for the rich semantics it provides which is well suited for rule definitions. We will use 650 scenes

for training and 200 scenes for validation. Details in experiment setup, implementation and hyper-parameters used are provided in the Appendix C.

**Rules used.** We use the following two rules to describe behaviors of a safe ado agent. Our goal is to generate ego plans which avoid ado agents deemed risky in the sense that they violate these rules.

$$\phi_1 = G^-_{[0,H)} (\text{driveNearLane} \wedge \text{keepSafeCarDistFromEgo})) \tag{6}$$

$$\phi_2 = G^-_{[0,H)} (\text{egoInIntersection} \rightarrow (\text{farFromIntersection} \vee \text{driveSlowly})) \tag{7}$$

where $H$ is the tracking horizon (we use $H = 4$). $\phi_1$ describes that a safe ado vehicle should "*always* drive near the center lane *and* keep a safe distance from the ego vehicle." $\phi_2$ expresses that at intersections "*always* if the ego vehicle is in the intersection *implies* that a safe ado vehicle should either be far from the intersection *or* drive slowly." The final rule $\phi = \phi_1 \wedge \phi_2$ takes the form of a conjunction of a set of sub-rules. Predicate and parameter definitions can be found in Appendix B.

**Methods of evaluation.** We evaluate our method and comparison cases in terms of optimality (time to reach goal position) and safety (minimum distance to nearby vehicles). Within the dataset, we will set the human ego vehicle's start and end positions as the initial and goal positions. Optimality is measured as the time to travel from the initial position to within a distance to the goal position averaged over the validation set. Safety is measured as the minimum distance to nearby ado vehicles in a scene averaged over the validation set. We perform two experiments: 1) during evaluation, we control the ego vehicle with our learned planner, and the ado vehicles move according to the trajectories recorded in the dataset and time is synchronized; and 2) we re-run the evaluation in a more realistic setting using ado vehicles that react to the ego according to an intelligent driver model (see Appendix G). In each of our evaluations, the ego vehicle can navigate in an environment with realistic human or reactive ado vehicles.

We use four methods for comparison. *LogicRiskNet-IOC* refers to the proposed method; *Human* refers the human driver in the dataset; *BC* refers to a behavior cloning agent; and *TrajOpt* refers to a trajectory optimization agent. Implementation details of *BC* and *TrajOpt* agents can be found in Appendix C.

**Results and discussions.** LogicRiskNet does not only output the final risk measure but can also output the risk of all intermediate sub-formulas without needing additional computation. This allows us to explain the reason of its decisions. Figure 2 shows an example of an unprotected right turn at an intersection and the vehicle that we are monitoring is highlighted by the yellow dash circle in Figures 2(a) and 2(c). Here we are showing two consecutive steps in the scenes. Looking at the top plot in Figure 2(b) we can see that this vehicle is labelled as risky (a negative risk value denotes

---

**Algorithm 2** LogicRiskNet Guided LfD

1: **$Inputs$**: parameteric ptSTL formula $\phi^\xi$; parameter learning rate $\gamma^\phi$; demonstration dataset $\mathcal{D}$
2: $\xi \leftarrow \xi^{init},\ \beta^{\boldsymbol{u}} \leftarrow \beta^{\boldsymbol{u},\ init},\ \beta^{\mathcal{N}} \leftarrow \beta^{\mathcal{N},\ init}$
3: **for** i=1 ... N **do**
4: $\quad A_{f^\phi_{\mathcal{D}}} = \frac{1}{|\mathcal{D}|} \sum_{d=0}^{|\mathcal{D}|} \left( f^\phi \left( \boldsymbol{h}^d_H, \boldsymbol{\omega}^d_T \mid \xi \right) + f^{\boldsymbol{u}} \left( \boldsymbol{u}^d_T \right) + f^{\mathcal{N}} \left( \boldsymbol{\omega}^d_T,\ \mathcal{N}^d_T \right) \right)$
5: $\quad S_{\tilde{f}^\phi} = 0$
6: $\quad$ **for** j=1 ... $|\mathcal{D}|$ **do**
7: $\quad\quad \tilde{\boldsymbol{\omega}}_e, \tilde{f}^\phi, \tilde{f}^{\boldsymbol{u}}, \tilde{f}^{\mathcal{N}} = \texttt{TrajectoryOptimization} \left( \tilde{\boldsymbol{u}}^{init}_T,\ \boldsymbol{x}^{d,0}_e,\ \boldsymbol{h}^d_H,\ \mathcal{N}^d_T, \beta^{\boldsymbol{u}},\ \beta^{\mathcal{N}},\ \xi \right)$
8: $\quad\quad S_{\tilde{f}^\phi} += \tilde{f}^\phi + \tilde{f}^{\boldsymbol{u}} + \tilde{f}^{\mathcal{N}}$
9: $\quad$ **end for**
10: $\quad A_{\tilde{f}^\phi} = \frac{1}{|\mathcal{D}|} S_{\tilde{f}^\phi}$
11: $\quad L^\phi = (A_{f^\phi_{\mathcal{D}}} - A_{\tilde{f}^\phi})^2$
12: $\quad \left( \xi,\ \beta^{\boldsymbol{u}},\ \beta^{\mathcal{N}} \right) \leftarrow \left( \xi,\ \beta^{\boldsymbol{u}},\ \beta^{\mathcal{N}} \right) - \gamma^\phi \nabla_{\xi,\ \beta^{\boldsymbol{u}},\ \beta^{\mathcal{N}}} L^\phi$
13: **end for**
14: **return** $\xi,\ \beta^{\boldsymbol{u}},\ \beta^{\mathcal{N}}$

---

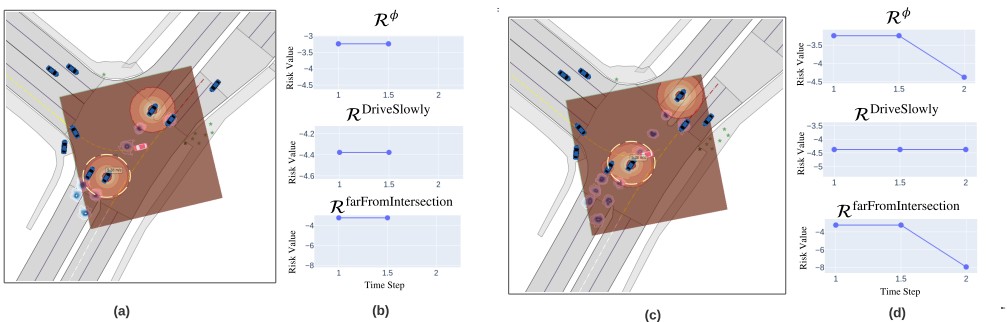

**Figure 2: LogicRiskNet monitor results.** (**a**) Monitoring the highlighted ado vehicle at time 1.5 sec. (**b**) Risk traces for $\phi$ and selected sub-formulas. (**c**) Monitoring at time 2 sec. (**d**) Corresponding risk traces.

**Table 1:** Performance comparison results

| | Time to goal | | | | Safety distance | | | | Goal% |
|---|---|---|---|---|---|---|---|---|---|
| | **min** | **mean** | **max** | **90th** | **min** | **mean** | **max** | **90th** | |
| BC | 13.21 | 19.98 | 20.0 | 20.0 | 0.34 | 4.22 | 10.86 | 8.66 | 37% |
| TrajOpt | 16.92 | 18.50 | 20.0 | 19.76 | 3.89 | 6.37 | 12.43 | 8.34 | 87% |
| RiskLogicNet-IOC | 15.31 | 17.35 | 20.0 | 19.0 | 2.72 | 3.98 | 9.01 | 6.12 | 93% |
| Human | 19.0 | 19.75 | 20.0 | 20.0 | 3.03 | 4.62 | 9.45 | 6.84 | 100% |

violation of the safety rule $\phi$.) To gain more insights on why this is the case, we can look at the risk values of sub-formulas. The middle and bottom plot shows that this vehicle is neither driving slowly nor far away from the intersection. Figure 2(d) shows that after one time-step the risk for the monitored vehicle has increased (risk value becomes more negative), and the cause is not because the vehicle has sped up but because it has driven closer to the center of the intersection. The structure of the entire LogicRiskNet and the monitoring traces for all sub-formulas are shown in Appendix D.

Table 1 shows comparison results in terms of the time it takes to reach the goal and the minimum distance to neighboring vehicles during this navigation. An aggressive agent will achieve a low time-to-goal (drive faster) as well as a small safety distance. The %Reach goal measures the percentage of scenes in the validation set the corresponding agent is able to reach the goal state. Failure is called if the agent does not reach the goal in the duration of the scene. All results are averaged over scenes in the validation set (each scene is $\sim$20 seconds). From the table we can observe that *LogicRiskNet-IOC* achieves the most similar behavior to *Human* with the former slightly more aggressive. In comparison, *TrajOpt* is much more conservative. This is because it treats all agents as the same collision object that is needs to avoid whereas *LogicRiskNet-IOC* agent will place less emphasis on low risk agents (i.e. vehicles that are stopped). This gives the latter more free space to plan and also behaves more human-like (as humans place different amount of risk on vehicles based on their past behaviors). *BC* agent has a wide span of behaviors but mainly suffers from distribution shift [17] shown by its low success rate at reaching the goal. While we use non-reactive agents in this evaluation, in Appendix G we also extend it to a more realistic simulated environment.

Once the network parameters $\xi$ and the objective parameters $\beta$ are learned, we obtain a system that behaves similar to the human demonstrator. In practice, this may not be satisfactory and the user may wish to further fine tune the system. We conduct a study to see how varying the magnitude of the risk feature in Equation (4) changes the behavior of the ego agent. In Figure 3, the x-axis represents a risk coefficient multiplied to $\tilde{f}^{\phi}$ on line 6 in Algorithm 1. We run Algorithm 1 across all scenes in the validation set and record the time-to-goal and safety distance. Figure 3 shows that with the increasing importance placed on the risk term, the ego vehicle becomes more conservative (driving slower and keeping a further distance to ado vehicles). This finding allows the user to control the ego vehicle to strike a desired balance between risk-averse and risk-seeking behavior.

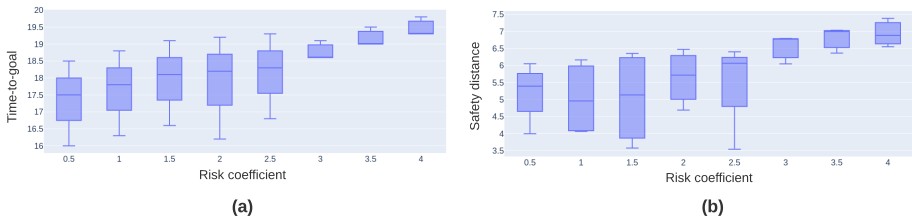

**Figure 3: Controlling risk management behaviors** by tuning risk feature coefficients. Both (**a**) Time-to-goal and (**b**) safety distance increases with the risk coefficient indicating that the ego vehicle becomes more conservative as we tune up this coefficient.

## 6 Related Work

*Risk and uncertainty aware policy learning* has been well-studied in the context of reinforcement learning. There have been several comprehensive surveys on this topic; [1] provide a general survey, while [2] provide greater focus on autonomous driving (AD). In general, common risk metrics such as conditional value at risk, mean-variance, worst-case analysis, etc. have been used as either an auxiliary loss in the objective function or as a set of constraints. In many robotic and driving tasks, however, RL is not readily applicable given its exploration requirement. Imitation learning and offline RL helps with addressing this problem. Specifically, in [3], kernelized movement primitives are used to estimate uncertainties in the demonstrations in finding optimal gains in a controller. The authors of [4] use offline distributional RL to learn a policy that is averse to conditional value at risk. Our work, in contrast, aims to learn from demonstrations to learn a human's notion of risk. It does not require exploration beyond the demonstrations, yet it is able to generalize well to new scenarios, and has the capacity to learn rare situations such as rule violations and near collisions.

*Temporal logic (TL)-guided policy learning* is an area that we take have taken inspiration from. In this area, TL is often used to specify the ego agent's desired high-level behavior and used to generate rewards. The authors of [18, 19, 20] provide surveys of recent work on the use of TL in RL. The exploration problem still exists in these methods. To address these challenges, the authors of [21, 22] learn finite state automata from demonstration and use them to guide planning with the value iteration network which avoids exploration. It can sometimes be tedious to manually design a TL formula that yields satisfying behaviors. Work has been done to make components of the formula learnable from data. In [23], the authors propose learning linear temporal logic (LTL) formulas from demonstrations. Given the close relationship between TL and automaton[24], the authors of [25] propose a method that learns reward machines (an automata-like reward presentation) from demonstrations. A shortcoming of these methods is that the LTL that they use operates on propositions (binary variables with values true or false). Unlike our STL-based approach, these approaches require discrete state and action spaces and require the demonstrations themselves to have the same discrete representations. We provide additional discussions on related work in risk-aware formal synthesis in Appendix H.

## 7 Conclusion

In this work, we introduced the LogicRiskNet - an expressive and differentiable risk representation based on temporal logic descriptions of risky agent behaviors. We show that with a demonstration dataset we are able to learn risk parameters that results in trajectories similar to the demonstrator. We also show that given the structure of LogicRiskNet we can monitor and explain why an agent is risky without additional computation. Given our choice of the IOC framework and the interpretability of our method, users are able to tune the objective function post-training to obtain the desired level of risk-averse behaviors.

**Acknowledgments**

This work is supported by the Toyota Research Institute (TRI). This article solely reflects the opinions and conclusions of its authors and not TRI, Toyaota, or any other Toyota entity. Their support is gratefully acknowledged. We also thank Nvidia for providing computation resources.

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
