# OpenReview forum: "Learning A Risk-Aware Trajectory Planner From Demonstrations Using Logic Monitor"
_robot-learning.org/CoRL/2021/Conference — CoRL2021 Poster_

### Official Review · Reviewer_gdH3 · 2021-07-23

**Originality:** Good
**Technical Quality:** Very Good
**Clarity Of Presentation:** Good
**Impact:** 4

**Recommendation:**

Weak Accept: I recommend accepting the paper, but will not argue for my recommendation if the majority of other reviewers have a different opinion.

**Summary:**

This work presents an architecture for learning to predict risk associated with other agents in a multi-agent setting (with focus on autonomous driving) in order to produce a risk metric to be used in planning. The architecture is based on a differentiable approximation of a temporal logic description of risks. The method takes a set of logic formulas which describe safe behaviour in terms of features, and is trained via imitation learning to produce risk management behaviour similar to an expert demonstrator. The learned risk model is interpretable. The method is compared against the human expert and 2 other approaches in an autonomous driving simulation, and is closer in performance to the expert driver than the existing approaches.

**Issues:**

- Line 17 - turning across traffic rather than turning left may be clearer for international readers
- Line 38 - should be ‘risk-averse’, not ‘risk-adverse’; same on line 39, 50, 238
- Would be good to briefly introduce 'ego'/'ado' terminology as this doesn't appear to be commonly used in other work
- Possible inconsistent use of \omega}_T between equation 2 and equation 3 - is the same symbol representing expert trajectories in one equation vs agent generated trajectories in the other?

**Reviewer Expertise:**

Fair: Some knowledge of the area

**Strengths And Weaknesses:**

Strengths:
- Novel and interesting approach
- Applicable to a number of important robotics problems
- Results show a good improvement compared to existing methods

Weaknesses:
- Some parts of explanation could be made clearer

**Summary Of Recommendation:**

The quality of the approach and results appear to be good. Presentation clarity is mostly good, with a few areas where clarity could be improved which should be revisited.

---

> ### Author Response · Authors · 2021-08-29
> **Response to Reviewer gdH3**
>
> We thank the reviewer for your comments and for pointing out some inconsistencies and clarifications. We agree with these comments and have modified the paper accordingly. In the revised paper, we have highlighted the modifications (other than typos) in blue. We use $\omega_T$ to represent expert trajectories and $\tilde{\omega}_T$ to represent agent generated trajectories. This is discussed at the end of Problem 1.
>
> Regarding clarity of the paper, we have revised sections 2, 3 and 4 of the paper to clarify terminology, and to better explain the major concepts relating to the RiskLogicNet. We have also added additional examples in Appendix F.

---

### Official Review · Reviewer_Qizq · 2021-07-23

**Originality:** Very Good
**Technical Quality:** Good
**Clarity Of Presentation:** Good
**Impact:** 3

**Recommendation:**

Weak Accept: I recommend accepting the paper, but will not argue for my recommendation if the majority of other reviewers have a different opinion.

**Summary:**

In this paper, the authors introduce LogicRiskNet, a form of differentiable temporal logic.
The parameters of LogicRiskNet can be learned from a set of demonstrations.

LogicRiskNet incorporates a measure of the risk connected to the agent's behavior. Furthermore, the logic encoded by LogicRiskNetwork, allows to interpret and explain the agent's choices.



**Issues:**

As already remarked, I believe that the authors must better clarify
- what is the risk measure considered (and why it is called risk measure). Does it only apply to autonomous driving or to more general set-up?
- how could one generalize the algorithm to a wider scenario? Is it possible? (for me it is fair that the algorithm focuses only on autonomous driving, but then the motivation of the work should be better specified).

**Reviewer Expertise:**

Poor: Limited knowledge of the area

**Strengths And Weaknesses:**

NOTE: as I declared in the "Reviewer Expertise", my knowledge in the area is limited (I do not have direct experience with temporal logic, and with inverse reinforcement learning or control).
I think that I understand the general idea of the work and the logic that connects each step. In this sense, I think that the paper is well written. I am however uncertain about the usage of the terminology "risk", and it seems that the algorithm is applicable to a small set of tasks, and this is not reflected in the abstract.

__Strenghts__

The paper presents a method to mimic the behavior of a demonstrator based on his risk awareness.
The model provided relies on logic and therefore is interpretable. Furthermore, the model is made differentiable by replacing classic $\max$ and $\min$ with their soft version. The differentiability allows the model to be trained with classic gradient-based techniques.

__Weakness__

I am not sure that I understood correctly some parts on of the paper. Risk, according to my knowledge, can be seen as a modification of an original distribution $p$, so to give more "probability" to rare "bad" events (risk-averse) or rare "good" events (risk-seeking). Common measures are
- Value at risk (VaR) (returns the value at a given quantile)
- Average value at risk (AVaR) (returns the average of the distribution from 0 to x quantile)
- Entropic value at risk (EVaR) (returns the average under a modified distribution $q$ such that $\max_q \mathbb{E}[X]$ with $D_{\text{KL}}(q||p) <= const$).

Equation 4 presents a sort of "potential field", and it is related, if I understand correctly, to the distance of two vehicles. The risk measure presented, does not seem to be really a risk measure. Perhaps the name is deceiving.
Does this "risk" measure apply therefore only to autonomous driving or to more general problems?



In general, the paper seems to be focused only on the particular case described in the example in Figure 1. From the abstract, it seems that the approach is far more general than the autonomous driving example, and autonomous driving has only been used to test it. However, from the paper, it seems that the algorithm is appositely designed for autonomous driving.






**Summary Of Recommendation:**

In general, the paper seems reasonable and well done. However, the writing is a bit deceiving.
It is not clear what the risk measure is. And it is not clear if the method is specifically thought of for autonomous driving.
If so, I'd recommend the authors specify that more clearly in the title and perhaps in the title too.

UPDATE
==

I want to thank the authors for their response. I appreciated the changes in the text. In my opinion, the exposition can be still enhanced, and I still think that the authors could have explained better how to generalize their approach, (i.e., lines 53 and 54 tell us that you can generalize, but not how you do that).
In general, my evaluation of the paper is positive, and I keep my score unchanged.

---

> ### Author Response · Authors · 2021-08-29
> **Response to Reviewer Qizq**
>
> We thank the reviewer for your comments. Regarding the comment that the risk measure presented does not seem to be really a risk measure,  as explained in the paragraph before eq. 3, we define the “risk measure” as a measure that maps a random variable to a real value (denoted $\rho$ in the paper).  You are correct in pointing out that this can take on any variety of forms: in our case the measure must be “coherent”, which includes expected value, mean-variance, value-at-risk, and conditional value-at-risk.  In our setting, the risk measure is applied to the robustness value for the formula, and uses the same treatment as in [12] and [25].  In our experiments, we use the expected value risk measure. The risk measure in eq. 3 is then used to scale the “potential field” as described in eq. 4. Please refer to Appendix F for a concrete example.
>
> We agree with the reviewer that the definition of risk, and the discussion surrounding its application to STL formulas, were too concise.  We have improved the discussion in the revision, and we hope this addresses the reviewer’s concerns.
>
> Regarding the reviewer’s question about whether the approach can be applied to domains other than autonomous driving, we expect that it can apply equally well to other robotic and cyber-physical domains calling for interactions with humans.  One modification to our approach is that different features will be required to form the IOC cost function (eq. 5), which, in our examples, are tailored to the autonomous driving domain.  The generality of our approach is that, as long as the domain is (partly) governed by a set of rules and a set of demonstration data with naturalistic human behavior can be obtained, the LogicRiskNet and IOC weights can be trained to output trajectory plans that resemble the demonstrators’ risk management behaviors.  We have clarified these points in the revision in Sections 3 and 4, as well as adding a discussion paragraph in Appendix E (highlighted in blue).

---

### Official Review · Reviewer_c3zR · 2021-07-23

**Originality:** Good
**Technical Quality:** Good
**Clarity Of Presentation:** Fair
**Impact:** 3

**Recommendation:**

Weak Accept: I recommend accepting the paper, but will not argue for my recommendation if the majority of other reviewers have a different opinion.

**Summary:**

This paper proposes a framework to learn a motion planner (a parametrized distribution of trajectories) that takes into account the risk ok other agents. The proposed approach is to design a risk monitor that can process previous trajectories and compute a risk score to each of the ado vehicles. The monitor is then used to learn the motion planner. The proposed framework is then evaluated using the NuScenes dataset.

**Issues:**

- The authors should provide an empirical evidence of why historical data provide a good predictor of the future risk. The authors may use the same dataset used in evaluation or any other dataset to provide such evidence.

**Reviewer Expertise:**

Very good: Comprehensive knowledge of the area

**Strengths And Weaknesses:**

Strengths:
- The motivation behind using risk as a factor in motion planning is interesting.
- The use of ptSTL leads to a formal framework to reason about and quantify risk

Weaknesses:
- The paper lacks several technical details which makes it hard to follow. In particular, the discussion on LogicRiskNet in Section 4 is vague. Examples are "If we use softmax approximations .... we can construct a computation graph ....". It is not clear what the authors are referring to in this paragraph. A more detailed construction is needed to understand the constructions that are verbosely described in this Section.
- One hidden assumption is that previous behavior of ado cars is a good predictor of their risk in the future. The paper failed to explain why this is indeed the case.
- The evaluation section is based on simple hand-picked risk rules. It is not clear how the performance of the proposed framework is affected by the choice of these rules.

**Summary Of Recommendation:**

The idea of using ptSTL to capture risk and design a risk monitor is interesting. Nevertheless, the paper is lacking deep technical discussion about several of the key components of the framework.

---

> ### Author Response · Authors · 2021-08-29
> **Response to Reviewer c3zR**
>
> We thank the reviewer for your thoughtful comments. The following are our attempts to address your concerns.
>
> Thank you for pointing out the lack in explaining the LogicRiskNet in Section 4. We have added an example of its construction in Appendix F (highlighted in blue) and referred to it in Section 4.
>
> In our experiments, we use the past 2 seconds of observations to calculate the risk factor and use it to generate a trajectory plan 3 seconds into the future. You are right in that during this process, we assume the risk of the agent remains constant for the next 3 seconds. During deployment, we execute this plan for only 1 timestep and then re-plan (in a receding horizon fashion), therefore, the risk of the agent is in fact re-evaluated at each timestep. We agree that it is helpful to investigate if historical data provides a good indicator of future risks. Therefore, we have added a new set of results to Appendix E (highlighted in blue, starting from the second paragraph). In this experiment, we calculate the risk values along the trajectories for all agents in the dataset (we refer to the result as the agents’ risk trajectory). Then, for every point on the risk trajectory, we take 3 seconds of its future risk trajectory and calculate the max percentage change with respect to the current risk value. The results show that under our given set of rules, the max future risk change is around 18 percent for all vehicles in the dataset. Given the fact that we re-evaluate risks and re-plan at every step, the assumption that past behavior is indicative of short term future risk should not affect the overall performance.
>
> The rules are the design factors in our method and the performance of the planner will vary according to how they are crafted (similar to the design of any objective function). There are references that can provide good starting points. In our case, in addition to common sense driving, we also referred to the California driver handbook (https://www.dmv.ca.gov/portal/handbook/california-driver-handbook/laws-and-rules-of-the-road) for safe interaction navigation.

---

### Official Review · Reviewer_3SPR · 2021-07-24

**Originality:** Very Good
**Technical Quality:** Good
**Clarity Of Presentation:** Very Good
**Impact:** 3

**Recommendation:**

Weak Reject: I recommend rejecting the paper, but will not argue for my recommendation if the majority of other reviewers have a different opinion.

**Summary:**

This paper introduced a risk-aware motion planner for autonomous vehicles that learns from expert demonstrations. The learning structure was called LogicRiskNet. In particular, the learned trajectories are guided by temporal logic rules. In addition to evaluating the risks of surrounding vehicles, the learned structure is able to explain why a surrounding vehicle is in a risky situation. Finally, the developed approach was evaluated using a dataset called NuScenes, which was selected due to its rich semantics. The proposed method was compared with a few baselines, including naive behavior cloning, and optimization methods. Results showed that the developed approach produced high success rate, and shorter time to navigate to the goal.

**Issues:**

See above.

**Reviewer Expertise:**

Very good: Comprehensive knowledge of the area

**Strengths And Weaknesses:**

The main concern is that autonomous driving is a highly interactive domain, but the paper used prerecorded data to simulate the interactions. The differentiable logic-based risk metric is fine, because it doesn't involve any changes to the world (simulated or not). It's becoming problematic when evaluating the proposed control method (risk-adverse and risk-seeking behaviors). It does not make sense to evaluate the performance of a vehicle assuming the surrounding vehicles are not responsive at all. This issue was briefly mentioned in the "Methods of evaluation" paragraph, but it's pointing to future work to address this issue. It's unclear why simulators like CARLA was not used in the paper.

The other concern is that Sections 2 and 3 were not presented in a way that's friendly to readers who are not very familiar with the relevant mathematics. The reviewer knows basic knowledge representation of AI, and has some experience about LTL, but still find it difficult to follow the symbols and equations. It's good that there are transition sentences that summarize the technical details, which helped the reviewer follow the whole paper. Still, it will be good if the paper can be presented in a way that's mathematically less heavy.

There were only two formulas used in the evaluation, though the developed approach is supposed to be generally applicable to logic formulas. Also, the explanation part of the work was not evaluated.

A few minor issues:

 - Figures show up before their references in the body of the paper. For instance, Figure 2 is on page 6, but it was not mentioned until page 7. Figure 1 is on page 2, and its first mention is on page 4.

 - Font size in figures is very small, and the embedded text is hardly readable.

 - Please explain what "ado agent" means the first time it's used. The reviewer couldn't figure it out until reading Figure 1.

 - The paper has many sentence statements that start with "The authors of [X]..." In general, a reference is to support a statement, but should not be a statement itself.

**Summary Of Recommendation:**

The logic-guided risk evaluation part of the paper (LogicRiskNet) is well motivated, and makes a lot of sense. But the evaluation part has a control component, and was evaluated using a prerecorded dataset, which is questionable and not convincing.

---

> ### Author Response · Authors · 2021-08-29
> **Response to Reviewer 3SPR**
>
> We appreciate the reviewer’s comments and suggestions for improvements.  We believe each of these points are valid.  We respond to specific concerns below, which we hope addresses each of the concerns raised.
>
> On the concern that we are using a static dataset without interaction.  We acknowledge that this interactivity is required in order to perform a realistic evaluation (as opposed to a worst-case evaluation).  Rather than acknowledging this as future work (as the reviewer notes), we are actively working on including results with interactive traffic participants in this manuscript, and we anticipate having these results in the revised manuscript before the revision deadline (see below for further details).
>
> On the use of simulators (e.g. CARLA).  While it would be useful to have simulated agents in light of the above concerns regarding interactions, we believe that using a simulator such as CARLA would not further substantiate the experimental evaluations in Section 5.  Note that the aforementioned re-working of our experiments to include traffic interaction does not require a simulator, only simple models of reactive agents, as we do not train directly on images, only post-processed tracks.  Finally we note that the interactive traffic models we are now using behave similar to CARLA’s agents.
>
> On the clarity of Sections 2 and 3.  We agree with the reviewer that the section could be improved for clarity, especially the notation and background information.  We have re-worked these sections in the revision and included a concrete example in Appendix F, we have also addressed all the minor issues raised by the reviewer (note the changes appear in blue).
>
> We have posted a revised manuscript with a snapshot of the changes we made to address the motivations for using the formulas we did, acknowledge future work and clarify points brought up by other reviewers.  We hope to have an updated manuscript posted with the results with interactive traffic participants within two days’ time.

---

> > ### Author Response · Authors · 2021-08-30
> > **Follow up**
> >
> > To follow up on the additional experiments, in Appendix G,  we provide results where reactive agents are inserted into the environment during evaluation. For each evaluation scene, we insert a random number (from 1 - 3) of controlled ado agents at the beginning of the scene (positions randomly distributed on driveways near the ego agent) and that lives throughout the scene. The reactive agents exhibit lane following behavior and their speeds are controlled by the intelligent driver's model (IDM). This results in the behavior that if there are no vehicles in the vicinity of the controlled ado agents, they will follow the target lane at a constant speed, otherwise, they will speed up or slow down to avoid collision. The overall effect of inserting controlled ado agents is that the evaluation environment is more crowded than the original. Hence results show that safety distance and goal reaching percentage decreased whereas the time to goal increased. However, this decrease in performance is minimal for our method due to the fact that IDM agents are good at giving way to others and hence in general are safe agents.
> >
> > We are not sure what the reviewer means by “the explanation part of the work”. But if it refers to the explainability of our method, we provide results in Appendix D and Figure A3 where monitoring traces of the LogicRiskNet are plotted and we discuss how we can diagnose the reason for a risky agent given these traces.

---

### Meta-Review · Area_Chair_BNdp · 2021-08-12

**Recommendation:** Accept (Poster)
**Confidence:** 4

**Metareview:**

In this paper the authors, by leveraging tools from temporal logic, design a framework to learn risk-aware trajectory planners. The framework is evaluated on a real-world driving dataset.

In general, the reviewers agree that the proposed idea is novel and useful, and I concur with this assessment. However, they also point out some weaknesses, including some lack of details in the exposition of the technical approach and a somewhat weak experimental evaluation (e.g., the experiments only consider two, rather simple logical rules).  The authors should carefully address the reviewers' concerns in their rebuttal.

UPDATE POST DISCUSSION PHASE: I would like to thank the authors for their comments and clarifications during the discussion phase. The reviewers generally agree that this paper presents an interesting idea and provides a useful contribution. On the other hand, some aspects of this paper still need to be improved - in particular, in terms of its exposition clarity (for example, the authors should better discuss to what extent they leverage the methodology presented in [14]). The decision is to accept this paper, but the authors should carefully address all reviewers' comments in the final version of this paper, with a key focus on improving exposition clarity.

---

> ### Author Response · Authors · 2021-08-30
> **Response to Area Chair BNdp**
>
> We thank the area chair and reviewers for your thoughtful comments. We summarize here the main issues raised by the reviewers and our attempts to address them.
>
> - Our method is evaluated on the dataset without interactive agents
>
> We added IDM agents to the environment during evaluation and provide an additional set of results and discussions in Appendix G.
> Our method uses past agent behavior to predict future risks and lacks the evaluation that this assumption is valid.
> In our experiments, trajectory plans are generated by assuming the risks of all ado agents remain constant for 3 seconds in the future. We execute the plan in a receding horizon fashion and therefore the risks are in fact re-evaluated at each timestep. In addition, we provide results in Appendix E that show for all agents in the dataset, the maximum change of their risks in 3 seconds is around 18% (mean around 10%). Therefore, we think this assumption does not negatively impact the performance of our method.
>
> - Sections 2, 3, 4 are notation heavy and lack intuitive explanations.
>
> We have modified these sections with added explanations and examples. We have also added a concrete example of how LogicRiskNet is constructed from a formula in Appendix F.
>
> - The rules we use for evaluation are simple.
>
> Rules are the design elements of our method, like any objective functions of optimization problems, their craft affects the final performance. Sometimes simple and widely applicable rules work well whereas other times rules specific to certain scenarios will help in those domains. In this work, we have referred to the California driver handbook (for intersection navigation) as well as common sense driving for designing the rules. Our main focus is to introduce the rule-based risk IOC framework which is general and supports arbitrary ptSTL rules.
>
> We have also modified the paper addressing all the minor issues raised by the reviewers. Hopefully our efforts can serve to better support our ideas and arguments in the paper and improve the paper’s overall clarity and readability.

---

### Decision · Program_Chairs · 2021-09-13

**Decision:**

Accept (Poster)

**Comment:**

In this paper the authors, by leveraging tools from temporal logic, design a framework to learn risk-aware trajectory planners. The framework is evaluated on a real-world driving dataset.

In general, the reviewers agree that the proposed idea is novel and useful, and I concur with this assessment. However, they also point out some weaknesses, including some lack of details in the exposition of the technical approach and a somewhat weak experimental evaluation (e.g., the experiments only consider two, rather simple logical rules).  The authors should carefully address the reviewers' concerns in their rebuttal.

UPDATE POST DISCUSSION PHASE: I would like to thank the authors for their comments and clarifications during the discussion phase. The reviewers generally agree that this paper presents an interesting idea and provides a useful contribution. On the other hand, some aspects of this paper still need to be improved - in particular, in terms of its exposition clarity (for example, the authors should better discuss to what extent they leverage the methodology presented in [14]). The decision is to accept this paper, but the authors should carefully address all reviewers' comments in the final version of this paper, with a key focus on improving exposition clarity.